# Application of EMGB to Study Impacts of Public Green Space on Active Transport Behavior: Evidence from South Korea

**DOI:** 10.3390/ijerph19127459

**Published:** 2022-06-17

**Authors:** Myung Ja Kim, C. Michael Hall

**Affiliations:** 1The College of Hotel & Tourism Management, Kyung Hee University, Seoul 02447, Korea; 2Department of Management, Marketing and Entrepreneurship, University of Canterbury, Christchurch 8140, New Zealand; 3Geography Research Unit, University of Oulu, 90014 Oulu, Finland; 4Ekonomihögskolan, Linnéuniversitet, Universitetskajen, Landgången 6, 39182 Kalmar, Sweden; 5Department of Service Management and Service Studies, Campus Helsingborg, Lund University, 25108 Helsingborg, Sweden; 6Centre for Research and Innovation in Tourism, Taylor’s University, Subang Jaya 47500, Malaysia

**Keywords:** active transport, walking/cycling, public green space, motivation theory, goal-directed behavior, smart app

## Abstract

Public green spaces (e.g., parks, green trails, greenways) and motivations to engage in active transport are essential for encouraging walking and cycling. However, how these key factors influence walker and cyclist behavior is potentially being increasingly influenced by the use of smart apps, as they become more ubiquitous in everyday practices. To fill this research gap, this work creates and tests a theoretically integrated study framework grounded in an extended model of goal-directed behavior, including public green space and motivation with perceived usefulness of smart apps. In order to accomplish the purpose of this study, we conducted an online survey of Korean walkers (*n* = 325) and cyclists (*n* = 326) between 10 and 25 July 2021 and applied partial least squares, structural equation, and multi-group analysis to validate the research model. Results revealed that active transport users’ awareness of public green space positively influences attitude toward (γ = 0.163), as well as behavioral intention of (γ = 0.159), walking and cycling. Additionally, motivation (extrinsic and intrinsic) greatly influences attitude (γ = 0.539) and behavioral intention (γ = 0.535). Subjective norms (γ = 0.137) and positive (γ = 0.466) and negative anticipated emotions (γ = 0.225) have a significant impact on the desire that leads to behavioral intention. High and low perceived smart app usefulness also significantly moderates between public green space and attitude (*t*-value = 25.705), public green space and behavioral intention (*t*-value = 25.726), motivation and attitude (*t*-value = −25.561), and motivation and behavioral intention (*t*-value = −15.812). Consequently, the findings are useful to academics and practitioners by providing new knowledge and insights.

## 1. Introduction

Public green space plays many important roles, including improving mental and physical health, enabling personal and public well-being, and contributing to environmental and/or climate change mitigation goals [1,2,3]. The provision of green space enhances active transport use, such as walking and cycling, resulting in benefits for human and environmental health [4,5,6,7,8,9,10]. Studies of extrinsic and intrinsic motivations also help explain pedestrian and cyclist behaviors.

Researchers have had a long-standing interest in the theories of goal-directed behavior [11,12,13], with models of goal-directed behavior (MGB) often being applied to examine sustainable consumer behavior [14,15,16,17,18]. Research on active transport has found that an MGB greatly explains pedestrian and cyclist behaviors [19,20]. However, the nature of walking and cycling is changing, with there being greater use of smart app technology; consequently, research on smart apps for active transport has found that they can influence walkers’ and cyclists’ perceptions in a variety of ways [21,22,23,24]. Nevertheless, despite the importance of green space and motivation for engagement in active transport, research on public green space and extrinsic/intrinsic motivation has been neglected in terms of the role of smart apps for walking and cycling based on goal-directed behavior theory.

### 1.1. Theoretical Context

#### 1.1.1. Public Green Space Relevant to Walking and Cycling

Given its diverse and many social, economic, and environmental benefits, public green space, e.g., parks, gardens, forests, woods, greenbelts, and green spaces that are openly available to the public, is indispensable to society [3,25]. It is well recognized that urban green spaces have a positive influence on individuals’ health and well-being [2,26]. The availability of green space is associated with reduced adult mortality, improved social capital, lower stress, and improved air quality [1,27]. Positive impacts of green spaces on individual activity behavior (e.g., walking) and health have also been increasingly highlighted in terms of street design and planning [7,9,10].

Compared to areas with poor public green space and trail access, people residing in neighborhoods with high availability of public green space and cycle trails are more likely to walk and cycle [5,8]. The significance of walking in urban green space to health has also been highlighted, revealing the potential health benefits of supporting such behavior [4]. As well as encouraging physical activity, public green space can also help mitigate air pollution and urban heat island effects [6]. Therefore, this study considers public green space as an important public good that can encourage the active transport of walking and cycling and contribute to more sustainable urban environments. Thus, public green space is defined as parks, gardens, forests, woods, greenbelts, greenways, and green trails; such green space can improve mental and physical health, enable personal and public well-being, and contribute to environmental and/or climate change mitigation goals [4,5,6,7,8,9,10].

#### 1.1.2. Motivation Theory

Motivation can be defined as a desire, drive, wish, goal, and/or need, which is classified as being both extrinsic and intrinsic in nature [28]. Extrinsic motivation refers to behaviors and/or actions, which are a possible achievement of some outcome, whereas intrinsic motivation are behaviors, which engage in an action for the fulfillment or gratification inherent in the execution of the action [29,30]. Specifically, intrinsic motivation remains an important construct, reflecting the natural human propensity to learn and assimilate; however, extrinsic motivation may vary considerably in its relative autonomy and can thus either reflect external control or true self-regulation [31,32]. Intrinsic motivation is also found to be positively related to moderate-to-vigorous physical activity [33].

Researchers have been interested in motivations relevant to walking and/or cycling from a variety of disciplines and perspectives [34,35,36,37]. This is partly because the motivations, which persons have for active transport activities—walking trips, for example—are a strong indicator of the degree of satisfaction with the activity [35]. Extrinsic (fitness and appearance) and intrinsic (competence, enjoyment, and social) motivational impacts on participation in a walking campaign are diverse according to demographic characteristics [31]. Leisure benefits are positively affected by cycling motivation, and both cycling motivation and leisure benefits have a positive impact on well-being [37]. Rejón-Guardia et al. [33] found that the primary motivations for participation in mountain cycling are weather (physiological or perceived) and route signage, followed by utilitarian practices (cost of the trip) and intrinsic motivations, such as the enjoyment of being engaged in physical challenges.

Based on the literature, this research therefore regards motivation (intrinsic and extrinsic) as a vital variable in the context of walking or cycling.

#### 1.1.3. Models of Goal-Directed Behavior (MGB)

The theory of reasoned action (TRA) suggests that human attitudes as well as subjective norms can forecast intentions, applying beliefs, attitudes, intentions, and behaviors [38]. Subsequently, the theory of planned behavior (TPB) was developed to resolve the limitations of the TRA by dealing with issues of incomplete volitional control [39]. In other words, TPB emphasizes that human behavior is governed by human attitudes and subjective norms (social pressures) and a perceived sense of behavioral control [40]. TPB also describes behavior in terms of it being preceded by an intention to complete the predicted behavior, an element of the theory, which is found to be well supported by empirical evidence [39]. More recently, an MGB has been established by developing an extended TPB incorporating anticipated emotions and desire [40].

Theories of goal-directed behavior are regarded as providing a suitable conceptual framework for examining both routine and non-routine travel [13]. The MGB has been found to predict travel behavioral intention with respect to subjective norms, attitude, positive and negative anticipated emotion, desire, and perceived behavioral control [38,39,40], as well as by behavioral studies [41,42]. For example, during a flu pandemic, consumer travel intention was highly explained by an EMGB, which added key constructs of non-pharmaceutical interventions and perception of infectious diseases [43]. Differences in international travel behavior between males and females has also been significantly explained by the MGB, along with frequency of past behavior [44]. Consumer behavior has been substantially predicted by the goal-directed behavior, including key variables of motivations of extrinsic (usefulness) and intrinsic nature (enjoyment) [29,41,42]. In an international crisis setting, the EMGB well identified potential consumers’ behavioral intention by adding important constructs, such as physical, financial, privacy, and performance as perceived risks [45].

In sustainable travel research, studies have used an EMGB to understand behavior [16,17]. For instance, responsible travel behavior has been explained by the EMGB, incorporating critical factors of perceived ethics (economic, socio-cultural, and environmental concern) [15]. In crowdfunding settings, funder behavior has been forecast by an EMGB with the additional concept of co-creation [16]. Research has also shed light on crowdfunding of Sustainable Development Goals (SDGs)-related projects, with the inclusion of constructs of perceived risks (financial, privacy, and performance) [17]. Travel mode choice, cycling, consumer attitude and behavior in active travel, and public transport have also been predicted by the MGB [19]. Consumer attitude and behavior with respect to urban sustainable mobility was identified by the MGB, together with prescriptive and descriptive norms [20]. Grounded in such literature, this study applies the EMGB, incorporating key concepts of awareness of public green space as well as motivation as a formative second-order construct with extrinsic and intrinsic sub-factors.

### 1.2. Hypothesis Development

The object of the work is therefore to create and assess a conceptual framework based on an extended MGB (EMGB), adding key variables of awareness of public green space and extrinsic and intrinsic motivation, along with the moderating effect of perceived usefulness of smart apps. This study raises three research questions: Do awareness of public green space and motivation for using active transport influence attitude, desire, and behavioral intention to walk and cycle; does an EMGB model predict walkers’ and cyclists’ behavioral intention; and does the perceived usefulness of smart apps moderate between awareness, motivation, attitude, desire, and behavior? Given the importance of active transport and green space for public and environmental health, the findings of the work potentially offer valuable academic and managerial insights for research and action on active transport initiatives.

Active transport use in urban green space is well recognized as a stress reduction behavior and for having positive benefits for mental and public health [46,47,48,49]. Awareness can be defined as the state of being cognizant of something and, importantly, it is the capability to identify and notice, to sense, or to be conscious of actions, positions, and/or some information in an extensive range of behavioral intentions [50]. Cycling behavior in a travel context is influenced by cognitive (e.g., awareness), evaluative (e.g., attitude), and motivational processes (e.g., desire) [51]. In terms of green mobility, consumers’ awareness of consequences has a significant impact on their attitude that then influences intentions and behaviors [52]. In addition, eco-friendly knowledge positively influences ecological attitude relevant to green purchase behavior [53], implying that awareness can lead to attitude, desire, and behavioral intention. Environmentally friendly awareness has also been found to significantly influence green purchase intention [54]. From a sustainability perspective, the perceived economic, socio-cultural, environmental concerns (e.g., awareness) significantly influence attitude toward and behavioral intention in relation to responsible travel behavior [15]. Hence, we propose three hypotheses:

**Hypothesis** **1a** **(H1a).***Awareness of public green space has a positive effect on attitude toward walking and cycling*.

**Hypothesis** **1b** **(H1b).***Awareness of public green space has a positive effect on desire to walk and cycle*.

**Hypothesis** **1c** **(H1c).***Awareness of public green space has a positive effect on behavioral intention of walking and cycling*.

Intrinsic motivation impacts satisfaction related to attitude, which, in turn, influences supporting behavior [30]. Travel consumers’ extrinsic and intrinsic motivations have significant effects on attitude relevant to desire, leading to their behavioral intentions [29]. Consumers’ motivation influences their flow experience related to well-being and behavioral intention from a travel perspective [55]. Motivations to belong to social groups influence the desire of people to develop and maintain stable inter-personal relations [56]. In a green bed and breakfast context, consumers who are motivated by environmental friendliness are more likely to have a strong desire toward green experiential loyalty [57]. Intrinsic motivation is highly correlated to leisure-time physical activity, such as walking [58]. Individuals’ motivations for participating in walking trips are a strong predictor of their satisfaction level that significantly leads to a desire to walk [35]. Grounded in the literature review above, we posit three hypotheses:

**Hypothesis** **2a** **(H2a).***Motivation has a positive effect on attitude toward walking and cycling*.

**Hypothesis** **2b** **(H2b).***Motivation has a positive effect on the desire to walk and cycle*.

**Hypothesis** **2c** **(H2c).***Motivation has a positive effect on behavioral intention of walking and cycling*.

Perugini and Bagozzi [40] suggest that desires can be defined as the proximal cause of behavioral intentions and fully mediate between anticipated emotions, attitudes, subjective norms, perceived behavioral control, and intentions. These relationships are significant in travel environments [43,44,45]. In travel consumer behavior, associations with attitude, perceived behavioral control, negative anticipated emotion, positive anticipated emotion, desire, and intentions are positively significant [16]. In responsible travel contexts, relationships between attitude, subjective norms, perceived behavioral control, positive anticipated emotion, and negative anticipated emotion toward the desire relevant to behavioral intention are significant [15,19]. Subjective norms, perceived behavioral control, and positive and negative anticipated emotion have significant impacts on desire, which leads to behavioral intention in sustainable consumer behaviors [17,20]. In line with this literature, we suggest six hypotheses:

**Hypothesis** **3a** **(H3a).***Attitude has a positive effect on the desire to walk and cycle*.

**Hypothesis** **3b** **(H3b).***Subjective norms have a positive effect on the desire to walk and cycle*.

**Hypothesis** **3c** **(H3c).***Perceived behavioral control has a positive effect on the desire to walk and cycle*.

**Hypothesis** **3d** **(H3d).***Positive anticipated emotion has a positive effect on the desire to walk and cycle*.

**Hypothesis** **3e** **(H3e).***Negative anticipated emotion has a positive effect on the desire to walk and cycle*.

**Hypothesis** **4** **(H4).***Desire has a positive effect on the behavioral intention of walking and cycling*.

Increased use of multi-app-equipped smartphones is changing the way people walk and engage in physical activity and the way they use apps [59], which are increasingly incorporated into behavioral interventions to encourage walking [22]. A mobile phone app has been found to substantially increase personal mobility, including walking with respect to specific goal settings, self-monitoring, and/or responses [24]. Walking actions (such as unilateral stopping, turning, restarting) can be connected to actions displayed on a map app [24]. iPhone apps enabled reliable evaluations of walking activities [23]. Users of Google Maps and other apps are able to obtain safer routes for walking and crossing in urban areas [21]. These results suggest that individuals with high perceived usefulness of smart app for walking and cycling are potentially likely to possess different characteristics from individuals with low perceived usefulness of smart apps for active transport. Hence, we suggest the following hypothesis:

**Hypothesis** **5** **(H5a–H5f).***The high and low groups of perceived usefulness of smart apps have significantly different impacts on the relationships between public green space and attitude, public green space and desire, public green space and behavioral intention as well as motivation and attitude, motivation and desire, and motivation and behavioral intention of walking or cycling*.

The suggested relationships above are grounded in prior literature, applied in a range of different circumstances and locations. The study framework is presented in Figure 1.

## 2. Materials and Methods

### 2.1. Measurements

Due to measurement inaccuracies associated with single items, questionnaires were used after previously validated multiple measures were adapted for the context of this study [60]. In this study, the survey instrument originally included 44 questions and 11 concepts: awareness of public green space, extrinsic and intrinsic motivations, attitude, perceived behavioral control, subjective norms, positive anticipated emotion, desire, negative anticipated emotion, intentions, and perceived usefulness of smart apps.

Awareness of public green space was assessed with four items grounded in prior literature [1,5,6,15] (e.g., “I am interested in public green space for walking/cycling”). Extrinsic and intrinsic motivations were measured by four items each, drawing upon prior literature [15,30,31,32,33,34,37,55] (“Walking/cycling improves my personal health” and “Walking/cycling is enjoyable for me”).

Subjective norms, perceived behavioral control, attitude, positive and negative anticipated emotion, desire, and intentions of walking/cycling were assessed with four questions based on Kim and Hall [16,17] and Lee et al. [43] (e.g., “Walking/cycling is an affirmative behavior,” “Most people who are important to me think I should walk/cycle,” and “To increase my personal well-being, I am planning to walk/cycle”). The perceived usefulness of smart apps for walking and cycling was assessed with four questions drawn from previous literature [22,23,24,59,61] (e.g., “I believe that using smart apps for walking/cycling would enable me to accomplish walking/cycling better”).

Ranging between (1) strongly disagree and (7) strongly agree, a total of 44 questions were provided to respondents on a 7-point Likert-type scale. Seven-point Likert-type measures were used because they offer reliability as well as discriminant validity [62,63]. General characteristic questions associated with walking/cycling (i.e., participation types, reason for walking/cycling, comparison before COVID-19, using smart apps and types, personal safety, companions) were also added to the final survey. Seven items relevant to socio-demographics, such as gender, age, marital status, monthly household income, educational level, residential area, and occupation were also included.

### 2.2. Content Validity and Pilot and Pre-Test

To accomplish this study’s goals, the instrument was initially designed in English and then interpreted into the Korean version with three specialists suitably expert in the two languages reviewing the survey design. The Korean questionnaire was then back-interpreted into English to remedy any expression incongruities from English to Korean, resulting in the editing of both language versions of the survey [64].

For content validity, three experts in sustainable mobility performed a primary assessment of the measurements. As this stage, one item each of awareness of public green space, extrinsic motivation, and behavioral intention was added in order to better convey the intended meaning of each construct (i.e., “Green spaces are attractive for walking/cycling at any time of year,” “Walking/cycling improves public health,” and “To increase public well-being, I am planning to walk/cycle”). Several items were also slightly revised regarding public green space, extrinsic motivation, intrinsic motivation, and perceived usefulness of smart apps, so as to clarify questions. Three online survey specialists from the survey company used for this study also assessed whether the questionnaire could suitably evaluate walking and cycling. Subsequently, instructions, screen questions, socio-demographic variables, and general questions were developed to meet the requirements of digital survey systems. The questionnaire was also administered online to five Korean graduate students. The definitions of walking, cycling, active transport, leisure, tourism, and work were subsequently slightly reworded in light of their comments. A pre-test was then administered on 60 Korean residents who had walked or cycled for leisure, tourism, or work in the prior 12 months. Subsequently, as shown in Appendix A, three items relevant to respondent commitment to the quality of answers, time spent on answering the survey, and riding an e-cycle were added.

### 2.3. Data Collection

Due to their efficiency and efficacy, web survey services are increasingly used for online surveys, especially when large panels are available [65]. During the COVID-19 pandemic, online surveys are also an extremely appropriate data collection method for health and ethical reasons. The target respondents were Korean residents who were 18 years old and older and who walked or cycled. Since the initial outbreak of COVID-19 in January 2020 in Korea, travel has been strictly restricted under social distancing. Thus, this study did not specify a time period of experience with walking and cycling within a certain time. Based on resident registration demographics drawn from data provided by the Ministry of the Interior and Safety [66], a quota sampling technique was used in relation to the population age, gender, and residential area.

To obtain the data, a digital survey firm with the largest panel in Korea (approximately 1.5 million of Koreans) as of 4 September 2021 was used [67]. To ensure data quality, the firm used appropriate sampling measures, including: verification of participant identity; surveys finished too quickly were deleted; respondents not meeting the screening query were removed; every subject had a different question order to lessen answer bias; and subjects were asked to present the name of activity types among leisure, tourism, and work that they had most recently experienced by walking or cycling.

Data were collected through the online survey between 10 and 25 July 2021. An invitation detailing the study background and privacy measures was emailed to 4993 people for walking and 6191 for cycling subjects based upon a random sampling of 1,471,974 consumers on the survey company databases. Of the 2403 walking and 3003 cycling people who clicked on the invitation, 1528 (walking) and 1941 (cycling) individuals, respectively, looked further into it. All participants were required to specify whether they walked and cycled for leisure, tourism, and/or work (i.e., “Have you walked (cycled) for leisure-, tourism-, and/or work-related activities?”). At the beginning of this survey, definitions were presented. That is, in this study, “the term active transport refers to walking and biking; leisure activity means an activity chosen for pleasure, relaxation, or other emotional satisfaction, typically outside of work time, including daytrips; tourism is defined more generally as people traveling to and staying in places outside their usual home environment for no more than one consecutive year for leisure and no less than 24 h; walking can refer to walking for outdoor activity, leisure activity, recreation, exercise, hiking/tramping, and/or tourism-related activities; and biking can refer to biking for outdoor activity, leisure activity, recreation, exercise, sports, mountain biking, and/or tourism-related activities” (Appendix A). Only the 1258 panelists (492 cases for walking and 755 cases for cycling) who said “yes” to the screen questions were qualified to answer the questionnaire. Of those, 770 (walking: 370 cases, and cycling: 400 cases) validly completed the survey. After outliers and inappropriate respondents (e.g., completing the survey in less than 5 min) were deleted [68], 651 questionnaires were coded for the analysis, revealing a response rate of 51.7% (651/1258) using the American Association for Public Opinion Research criteria [69] (p. 58).

### 2.4. Data Analysis

This study conducted chi-square tests between the walking (325 cases) and cycling groups (326 cases) to understand whether they could be combined for analysis because the respondents were surveyed by separate walking and cycling groups. There is a statistically insignificant difference between walking and cycling behavior (Cochran, 1952) [70] according to Pearson chi-squared test (χ2 = 25.642, df = 27, *p*-value = 0.539). This study is therefore able to combine and analyze respondents from the walking and cycling groups to evaluate the proposed research model, since the null hypothesis that there are no differences between the two groups is true.

Partial least squares and structural equation (PLS-SEM) was used to evaluate the proposed research model. When assessing the first-order factors synchronously related to second-order factors, PLS-SEM is appropriate [71]. PLS-SEM is regarded as better than typical SEM, such as covariance-based approaches, when studies have non-normal distribution, small sample sizes, and/or complicated framework by multi-group analysis (MGA) [72]. Therefore, SmartPLS 3.3.3 was used to test the measurement as well as structural frameworks [73]. To validate the comparison between the two groups of high and low perceived usefulness of smart apps, MGA was used in accordance with PLS-SEM procedures [68,70], with moderating impacts tests as recommended by Chin et al. [74] and Keil et al. [75] (p. 315).

Furthermore, all tests (Harmon’s single-factor and marker variable approach) demonstrate that common method bias does not occur in this work (Appendix B).

## 3. Results

### 3.1. Sample Profile

Participants were almost equally distributed between male (49.3%) and female (50.7%). Around half of participants (44.9%) were in the 40–59 age bracket. The majority of the respondents had undergraduate or higher education (66.8%) and were married (61.9%). Most respondents lived in households with a monthly family income of over 4 million Korean Won (KRW) (USD 1 is equivalent to KRW 1142 as of 15 July 2021) and were engaged in full-time employment. More than a half of participants lived in the Seoul metropolitan regions.

The respondents answered all the questions, taking five minutes or more, as well as saying “yes” with respect to providing thoughtful and honest answers. The main reason to participate in walking and cycling is for physical well-being and health (67.0%). A little more than half of respondents engaged in walking and cycling more than or same as compared to before the pandemic (55.5%). A majority of respondents used smart apps for walking and cycling (64.7%). The most frequently used smart apps were ones that count the steps or measure the distance traveled (43.8%). During walking and cycling, just over two-fifths of respondents worried about their personal safety (40.2%). The majority of respondents (59.3%) usually walked or cycled alone. See more details in Appendix C.

### 3.2. Results of Measurement Model Testing

A shown in Table 1, confirmatory factor analysis (CFA) using all 47 indicators obtained factor loadings of over 0.7 [68]. Cronbach’s α, Rho_A as reliability coefficient, and the composite reliability of constructs were over 0.7, thus confirming internal consistency and reliability (Table 2) [76,77]. In confirming the convergent validity, the average variance extracted (AVE) of all constructs was over 0.5, and over 0.7 for the factor loadings of all indicators [71].

Based on Henseler et al. [78], the Heterotrait–Monotrait (HTMT) ratio was used to evaluate discriminant validity. The HTMT ratio with a cut-off value of below 0.90 is regarded as a more precise measure than the widely applied Fornell–Larcker criterion analysis for investigating discriminant validity [72,78]. The HTMT ratio in this study is below 0.9 of the cut-off, supporting discriminant validity. Furthermore, suitable relevance levels were achieved, as Q2 standards over zero (from 0.288 to 0.456) were confirmed for all endogenous variables [79,80]. The multicollinearity of variables was tested by employing the variance inflation factor (VIF). Since the outer VIF values were from 1.365 to 4.488, multicollinearity seems not to be a problem [68].

### 3.3. Results of Structural Model Testing

We utilized PLS-SEM to test the 14 relationships of five main hypotheses because the data indicated non-normal distributions by excess kurtosis (Table 2). The R^2^ (variance explained) shows attitude (42.5%), desire (58.6%), and behavioral intention (63.5%) (see Figure 2). To evaluate the sampling with non-normal distribution, the path coefficients, as well as the evaluated t-statistics of the relationships, were tested by employing PLS-SEM with bootstraps of 5000 re-samplings [71,74].

As demonstrated in Figure 2, awareness of public green space positively influences attitude (γ = 0.163, *t*-value = 3.617, *p* < 0.001) as well as behavioral intention (γ = 0.159, *t*-value = 4.703, *p* < 0.001). Motivation for walking and cycling highly influences attitude (γ = 0.539, *t*-value = 12.805, *p* < 0.001) and behavioral intention (γ = 0.535, *t*-value = 14.993, *p* < 0.001). Desire is significantly affected via subjective norms (γ = 0.137, *t*-value = 3.473, *p* < 0.001), positive anticipated emotion (γ = 0.446, *t*-value = 10.046, *p* < 0.001), and negative emotion (γ = 0.225, *t*-value = 5.997, *p* < 0.001). Behavior is significantly affected by desire (β = 0.219, *t*-value = 7.127, *p* < 0.001). Thus, H_1a_, H_1c_, H_2a_, H_2c_, H_3b_, H_3d_, H_3e_, and H_4_ were all supported. However, the relationships between awareness and desire (H_1b_: γ = −0.002, *t*-value = 0.056, *p* > 0.05), motivation and desire (H_2b_: γ = 0.083, *t*-value = 1.642, *p* > 0.05), attitude and desire (H_3a_: γ = 0.037, *t*-value = 0.857, *p* > 0.05), and between perceived behavioral control and desire (H_3c_: γ = 0.003, *t*-value = 0.080, *p* > 0.05) were insignificant, thus H_1b_, H_2b_, H_3a_, and H_3c_ were not supported. Furthermore, for the formative second-order construct, extrinsic motivation was more closely related to motivation (λ = 0.594, t = 42.106, *p* < 0.001) than intrinsic motivation (λ = 0.571, t = 35.084, *p* < 0.001).

In order to test the mediating roles of attitude as well as desire in the research framework, PLS-SEM bootstrapping by 5000 re-samples was utilized (Appendix D). Behavior was confirmed to be indirectly affected via subjective norms (γ = 0.032, *t*-value = 3.369, *p* < 0.001), positive anticipated emotion (γ = 0.116, *t*-value = 6.541, *p* < 0.001), and negative anticipated emotion (γ = 0.050, *t*-value = 4.798, *p* < 0.001). Thus, desire was confirmed to have mediating effects on the framework, while attitude had insignificant effects. The effect scales (f^2^), the f^2^ values (effect size), in this study were reported as ranging from 0.000 (no impact) to 0.394 (major impact) [81].

### 3.4. Moderating Effect of Perceived Usefulness of Smart Apps

The construct of perceived usefulness of smart apps was divided into high and low groups by the K-mean clustering method, with the average mean of four items (Appendix E). The high group of perceived usefulness of smart apps for walking and cycling had 433 cases with the mean score of 5.36, while the low perceived group had 218 participants with the mean value of 3.61. By using the PLS algorithms, each group was considered appropriate for MGA, letting the two groups be utilized for comparison [68]. As shown in Table 3, the high perceived group of smart app usefulness had greater impacts of awareness of public green space on the attitude and behavioral intention than their counterparts. In contrast, the low perceived participants of smart app usefulness had greater effects of motivation on the attitude and behavioral intention than their counterparts. Thus, H5 was partially supported.

## 4. Discussion

This work suggests that people with higher awareness of public green space (e.g., parks, gardens, forests, greenbelts, greenways, institutional green spaces) are more likely to have a positive attitude toward, as well as behavioral intention of participating in walking and cycling. Although such results may seem readily apparent, they actually extend previous literature on personal values and public green space in active transport in relation to attitude and behavior in urban areas [47]. The strong relationships between motivation and attitude as well as motivation and behavioral intention suggest that individuals with strong motivation for walking and cycling are more likely to have better attitude toward, as well as behavioral intention of participating in active transport behavior. These findings reinforce previous research on individuals’ motivations and satisfaction with respect to walking trips [35] and, perhaps more significantly, provide insights into more effective behavioral interventions that seek to encourage walking and cycling. This is particularly important, as extrinsic motivations are found to be more significant than intrinsic ones in terms of wanting to walk and cycle, extending previous findings on intrinsic and extrinsic motivations applied by the self-determination theory [31,32,33].

The positive influences of subjective norms, positive anticipated emotion, and negative anticipated emotion on the desire to walk and cycle are substantially significant. This work therefore expands previous findings on MGB in the context of active travel behavior for the travel mode choice cycle [19] and sustainable transport for cities [20]. In particular, the insignificant relationships between awareness and desire, motivation and desire, perceived behavioral control and desire, and attitude and desire suggest that people’s desires to walk and cycle are not influenced by awareness, motivation, perceived behavioral control, and attitude. A possible reason for these findings is that walking and cycling are essential to Korean residents who are dealing with the COVID-19 pandemic, so that even without the influences of awareness, motivation, attitude, and perceived control on desire, Koreans still continue to walk and cycle for leisure, tourism, and/or work. These findings are partially consistent with the findings of prior research on support for the Sustainable Development Goals (SDGs) in Korea [17].

With regard to the moderating effect of smart app usefulness, the stronger relationships between awareness of public green space and attitude as well as between awareness of public green space and behavioral intention from the high perceived group of smart app usefulness than the low group imply that a person with higher awareness of public green space for active transport tends to have stronger attitudes and behavioral intentions of walking and cycling if that person has a higher perceived usefulness of smart apps. These findings extend the literature on using apps for active transport in the context of smart cities [21,22]. On the other hand, the low perceived group of smart app usefulness has greater impacts on the relationships between motivation and attitude as well as motivation and behavioral intention compared to the high perceived group on smart app usefulness. These results are unexpected and contrary to our hypotheses and previous studies on fuzzy logic app for pedestrians [21]**.**

## 5. Conclusions

This work, based on applying consumer behavior theory to active transport, provides several theoretical contributions to the literature. First, the factors of awareness of public green space relevant to using active transport were found to have critical impacts on walking and cycling behavior. Second, adding extrinsic and intrinsic motivations reinforced the importance of motivation for walking and cycling in different contexts. Third, this work sheds light on potential roles in the EMGB in active transport consumer behavior. Finally, the significant moderating role of perceived usefulness of smart apps for active transport relevant to public green space, motivation, attitude, and behavioral intention potentially shows the importance of including the role of apps in assessments of active transport behavior for walkability and cyclability, given their increasing influence on consumer behavior in the active transport context.

This work also provides practical contributions to stakeholders. The effects of awareness of public green space on the attitude and behavioral intention reinforce the view that relevant public agencies should focus on highlighting the awareness of public green spaces, such as parks, green trails, and greenways, in order to increase positive attitude toward and behavior for active transport. The impacts of motivations on attitude and behavioral intention suggest that appropriately designed campaigns and interventions could boost peoples’ extrinsic and intrinsic motivations for walking and cycling, which, in turn, influence their attitude and behavior toward active mobility, resulting in improvements in the environment and their personal health. The positive influences of subjective norms and positive and negative anticipation emotions on desire to walk and cycle that this study identified may be particularly important variables for campaigns to focus on to encourage greater engagement, given the significant impact of desires on behavioral intentions.

The identification of the moderating effect of the perceived usefulness of smart apps for active transport also presents a potential new departure point for walking and cycling interventions that will potentially become even more important in the future, as cities seek to improve the sustainability of their environments and transport strategies. The results of this research highlight that when environmental, health, and transport agencies target consumers with high perceived usefulness of smart apps, then the general public’s awareness of urban green space should be stressed, drawing upon the significantly greater impact of city green space on the attitude and behavioral intention in the high group. Additionally, the results of this work suggest that when targeting people with low perceived usefulness of smart apps for active transport, then the general public’s motivation to walk and cycle should be emphasized according to the sufficiently greater effect of motivation on attitudes and intentions in the low perceived usefulness group.

## 6. Limitations and Future Research Directions

Although this work makes a number of contributions and has theoretical and practical implications, it also has some limitations that provide future study opportunities. First, this survey was only implemented in one country (South Korea) and during the COVID-19 pandemic (from 10 July to 25 July 2021), so the generalization of the findings needs to be carefully considered, with further research in other cultures and active transport environments required. Second, although this study outlines the importance of understanding walking and cycling behaviors in the specific context they occur, e.g., tourism, leisure, and work, more detailed examination of the different types of walking and cycling activities would be extremely valuable with respect to the purpose of such activities and their connection to other transport modes. Third, even though this study applied PLS-SEM and MGA, the adoption of other research methods, such as fuzzy-set qualitative comparative analysis (fsQCA) and in-depth interviews would also be valuable to understand more deeply the sequential relationships existing among the stated attitudes, behavioral intentions, and actual behaviors, particularly with smart app use. Furthermore, although this study sheds light on walking and cycling behavior in terms of green space, motivation, and MGB, a future study on the differences among tourism, leisure, and work activities would be valuable to better understand walking and cycling behavior.

## Figures and Tables

**Figure 1 ijerph-19-07459-f001:**
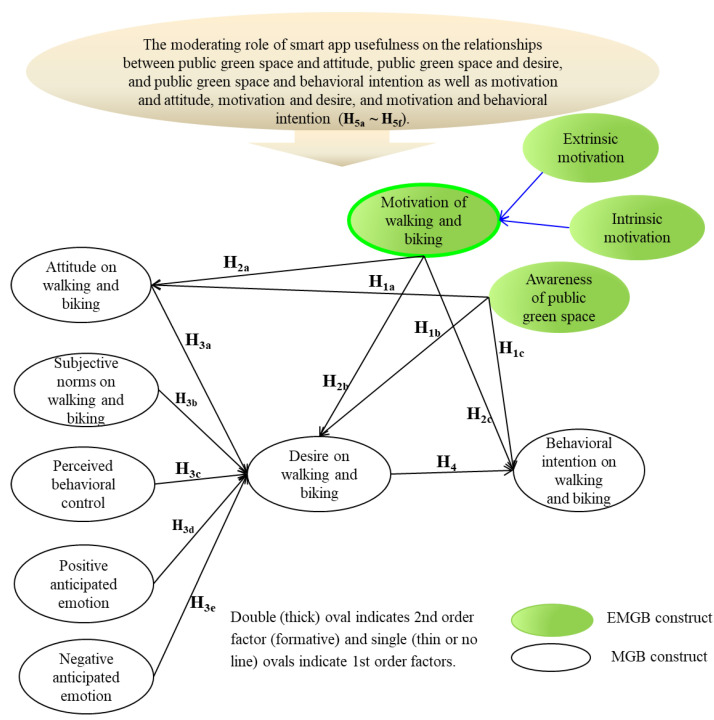
Proposed research model.

**Figure 2 ijerph-19-07459-f002:**
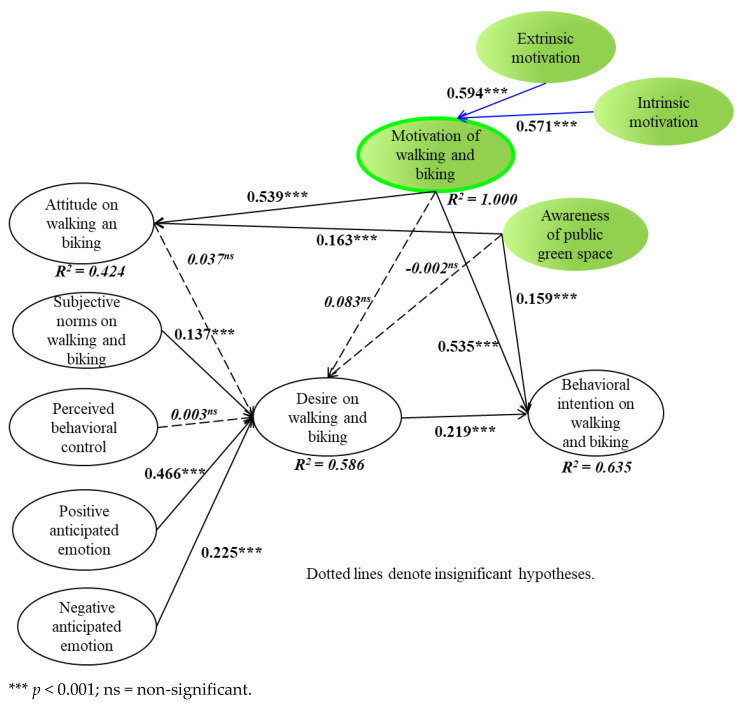
Results of path analysis.

**Table 1 ijerph-19-07459-t001:** Reliability and discriminant validity.

Construct	Heterotrait–Monotrait Ratio (<0.9)									
	1	2	3	4	5	6	7	8	9	10
1. Awareness of public green space										
2. Extrinsic motivation	0.548									
3. Intrinsic motivation	0.628	0.520								
4. Attitude	0.572	0.632	0.596							
5. Subjective norm	0.486	0.430	0.506	0.607						
6. Perceived behavioral control	0.502	0.522	0.517	0.574	0.399					
7. Positive anticipated emotion	0.594	0.568	0.860	0.618	0.545	0.502				
8. Negative anticipated emotion	0.409	0.248	0.377	0.456	0.519	0.306	0.415			
9. Desire	0.510	0.410	0.726	0.579	0.590	0.411	0.763	0.554		
10. Behavioral intention	0.677	0.732	0.754	0.689	0.603	0.503	0.755	0.442	0.694	
Cronbach’s alpha ≥ 0.7	0.856	0.899	0.909	0.849	0.884	0.766	0.927	0.942	0.914	0.852
Rho_A (reliability coefficient) ≥ 0.7	0.866	0.901	0.915	0.861	0.891	0.800	0.927	0.946	0.914	0.852
Composite reliability ≥ 0.7	0.898	0.926	0.937	0.899	0.920	0.846	0.948	0.958	0.939	0.894
AVE ≥ 0.5	0.638	0.717	0.787	0.691	0.742	0.581	0.82	0.852	0.795	0.629
Effect size (Q^2^) > 0	-	-	-	0.288	-	-	-	-	0.456	0.394

Note: -: Exogenous variables give effects to endogenous variables, so only endogenous variables have an effect size in causal modeling.

**Table 2 ijerph-19-07459-t002:** CFA on the measurements, descriptive statistics, and normal distribution.

Constructs	Factor Loading	*t*-Value	Mean	SD *	Kurtosis	Skewness	VIF **
**Awareness of public green space**							
1. I am interested in public green space for walking/cycling.	0.864	71.650	5.261	1.218	0.394	−0.653	2.685
2. I am aware of public green spaces for walking/cycling.	0.693	24.276	4.939	1.343	0.173	−0.628	1.486
3. I care about public green trails for walking/cycling.	0.866	75.457	5.427	1.164	0.640	−0.682	2.701
4. Public green spaces provide cool areas in which to walk/cycle when it is hot.	0.772	37.634	5.498	1.135	0.496	−0.727	1.782
5. Public green spaces are attractive for walking/cycling at any time of year.	0.787	35.861	5.450	1.154	0.503	−0.692	1.846
**Extrinsic motivation**							
1. Walking/cycling improves my personal health.	0.729	31.041	5.937	0.949	*1.597*	−0.987	1.562
2. Walking/cycling contributes to the environment.	0.903	70.813	5.533	1.120	0.460	−0.665	3.565
3. Walking/cycling contributes to mitigating climate change.	0.891	60.295	5.396	1.182	*1.003*	−0.789	3.558
4. Walking/cycling contributes to lowering air pollution.	0.895	94.862	5.512	1.212	0.896	−0.849	3.520
5. Walking/cycling improves public health.	0.802	42.136	5.144	1.118	0.120	−0.333	1.999
**Intrinsic motivation**							
1. Walking/cycling is enjoyable for me.	0.911	112.620	5.287	1.164	0.363	−0.551	3.366
2. Walking/cycling brings me self-satisfaction.	0.896	95.143	5.298	1.168	0.434	−0.550	2.887
3. Walking/cycling makes me happy.	0.915	132.335	5.169	1.178	0.199	−0.387	3.484
4. I walk for refreshment.	0.824	38.645	5.339	1.163	*1.025*	−0.763	2.037
**Attitude to active transport**							
1. Walking/cycling is an affirmative behavior.	0.881	90.493	5.690	1.009	*1.041*	−0.759	2.563
2. Walking/cycling is a beneficial behavior.	0.864	67.478	5.730	0.987	0.524	−0.648	2.418
3. Walking/cycling is an essential behavior.	0.713	26.581	4.763	1.428	−0.231	−0.397	1.498
4. Walking/cycling is a legitimate behavior.	0.857	67.214	5.255	1.100	0.275	−0.406	2.213
**Subjective norm on active transport**							
1. Most people who are important to me think I should walk/cycle.	0.860	65.546	4.293	1.409	−0.391	−0.223	2.445
2. Most people who are important to me would want me to walk/cycle.	0.897	82.026	4.516	1.331	−0.028	−0.288	3.016
3. Most people who are important to me support my walking/cycling.	0.819	37.580	4.980	1.244	0.633	−0.630	2.010
4. Most people who are important to me are proud that I go walking/cycling.	0.869	63.156	4.429	1.324	0.314	−0.345	2.187
**Perceived behavioral control**							
1. Walking/cycling or not is entirely up to me.	0.652	16.000	5.954	0.997	0.931	−0.990	1.412
2. I can walk/cycle whenever I want.	0.786	29.999	5.320	1.317	0.266	−0.798	1.683
3. I have the physical strength to walk/cycle.	0.750	23.633	5.696	1.025	0.139	−0.686	1.365
4. I have time to walk/cycle.	0.848	49.348	5.281	1.137	0.268	−0.542	1.696
**Positive anticipated emotion**							
1. If I walk/cycle, I will feel excited.	0.900	85.682	5.157	1.203	0.393	−0.521	3.098
2. If I walk/cycle, I will feel glad.	0.920	117.165	5.210	1.152	0.728	−0.556	3.782
3. If I walk/cycle, I will feel satisfied.	0.887	88.516	5.461	1.031	0.669	−0.582	2.747
4. If I walk/cycle, I will feel happy.	0.915	124.078	5.252	1.140	0.349	−0.449	3.576
**Negative anticipated emotion**							
1. If I cannot walk/cycle, I will be angry.	0.921	120.119	3.989	1.736	−0.848	−0.035	3.881
2. If I cannot walk/cycle, I will be disappointed.	0.932	157.084	4.607	1.695	−0.762	−0.390	4.200
3. If I cannot walk/cycle, I will be worried.	0.902	76.368	4.539	1.687	−0.869	−0.243	3.285
4. If I cannot walk/cycle, I will be sad.	0.937	174.769	4.458	1.750	−0.848	−0.278	4.488
**Desire to walk/cycle**							
1. I do want to walk/cycle.	0.847	67.976	5.167	1.197	0.712	−0.670	2.122
2. My desire to walk/cycle is passionate.	0.914	130.799	4.525	1.367	0.022	−0.431	3.495
3. I am enthusiastic about walking/cycling.	0.886	70.491	4.038	1.382	−0.267	−0.146	3.151
4. I am eager to walk/cycle.	0.916	128.309	4.258	1.397	−0.201	−0.308	3.827
**Behavioral intention in active transport**							
1. To increase my personal well-being, I am planning to walk/cycle.	0.785	42.778	5.301	1.099	0.746	−0.652	1.951
2. To improve my personal health, I will make an effort to walk/cycle.	0.740	26.692	5.605	1.093	*1.104*	−0.861	1.771
3. To mitigate climate change, I am willing to walk/cycle.	0.822	54.767	4.900	1.275	0.310	−0.588	2.574
4. To protect the environment, I do intend to walk/cycle.	0.822	50.032	4.693	1.372	−0.082	−0.387	2.795
5. To increase public well-being, I am planning to walk/cycle.	0.792	42.040	4.584	1.312	−0.037	−0.259	1.973
**Perceived usefulness of smart applications**							
1. I believe that using smart applications for walking/cycling would enable me to accomplish walking/cycling better.	0.897	89.006	4.628	1.244	0.067	−0.266	2.880
2. I believe that using smart applications for walking/cycling would improve my walking/cycling performance.	0.885	73.279	4.866	1.22	0.197	−0.358	2.871
3. I believe that using smart applications for walking/cycling would make it easier to do my walking/cycling.	0.861	60.945	4.567	1.329	−0.124	−0.304	2.238
4. I believe that using smart applications for walking/cycling would enhance my effectiveness in walking/cycling.	0.876	72.385	5.022	1.149	0.724	−0.493	2.664

Note: The italics indicate non-normal distribution. * Standard deviation. ** Variance inflation factor.

**Table 3 ijerph-19-07459-t003:** Moderating role of high and low perceived usefulness of smart apps.

H5	Path	High Group (A)	Low Group (B)	A–B	*t*-Value	*p*-Value	Hypothesis Test
H5a	Awareness of public green space → Attitude	0.206 ***	*0.076 ^ns^*	0.130	25.705	<0.001	Supported
H5b	Awareness of public green space → Desire	*0.017 ^ns^*	*−0.068 ^ns^*	0.085	20.052	ns	Not supported
H5c	Awareness of public green space → Behavioral intention	0.202 ***	*0.099 ^ns^*	0.102	25.726	<0.001	Supported
H5d	Motivation for walking/cycling → Attitude	0.470 ***	0.585 ***	−0.115	−25.561	<0.001	Supported
H5e	Motivation for walking/cycling → Desire	*0.086 ^ns^*	*0.027 ^ns^*	0.058	10.128	ns	Not supported
H5f	Motivation for walking/cycling → Behavioral intention	0.494 ***	0.561 ***	−0.067	−15.812	<0.001	Supported

*** *p* < 0.001; ns = non-significant. Since the two hypotheses in the high and low groups are insignificant, H5b and H5e are not supported.

## Data Availability

The data presented in this study are available on request from the first author. The data are not publicly available due to ethical reasons.

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
