# Peer review of "Application of EMGB to Study Impacts of Public Green Space on Active Transport Behavior: Evidence from South Korea"

_ijerph, 2022, doi:10.3390/ijerph19127459_

Round 1

Reviewer 1 Report

I think, the paper contains a quite interesting topic, but I have some suggestions:

  1. The paper is very long. Most of the tables can be replaced to the Appendix and the most important results, findings should be presented clearly in the chapters.
  2. There are a lot of references, which is very good, but it should be revised (e.g. reference should not appear in the conclusion chapter).

Author Response

Summary of Revisions and Responses to Reviewers’ Comments

Title: Do smart apps moderate the influence of public space and motivation on active transport? Evidence from South Korea

New title: Application of EMGB to study impacts of public green space on active transport behavior: Evidence from South Korea

Manuscript ID: ijerph-1708112.R1

Journal name: International Journal of Environmental Research and Public Health (IJERPH)

Reviewers' Comments to Author:
The three reviewers’ insightful comments are appreciated, and we have revised our manuscript based on the comments. We believe these comments have significantly improved the quality of our manuscript. Our revisions and responses to each of your comments are as follows (please also see all the blue text in our entire revised manuscript).

Reviewer 1:

English language and style

(x) Extensive editing of English language and style required

( ) Moderate English changes required

( ) English language and style are fine/minor spell check required

( ) I don't feel qualified to judge about the English language and style

Thank you so much for your thoughtful comment. As you suggested, we have conducted extensive editing of English language and style by professional and highly experienced editors and authors. Please refer to the blue text of the entire revised manuscript.

Yes      Can be improved        Must be improved       Not applicable

Does the introduction provide sufficient background and include all relevant references?

(x)       ( )        ( )        ( )

Are all the cited references relevant to the research?

(x)       ( )        ( )        ( )

Is the research design appropriate?

(x)       ( )        ( )        ( )

Are the methods adequately described?

(x)       ( )        ( )        ( )

Are the results clearly presented?

( )        (x)       ( )        ( )

Are the conclusions supported by the results?

( )        (x)       ( )        ( )

Comments and Suggestions for Authors

I think, the paper contains a quite interesting topic, but I have some suggestions:

We appreciate your valuable feedback and comments.

The paper is very long. Most of the tables can be replaced to the Appendix and the most important results, findings should be presented clearly in the chapters.

RESPONSE 1: With regard to your thoughtful suggestion, we have replaced three tables to the Appendices follows (please also refer to lines 817-825):

Appendix C. Demographic characteristic and general information of the entire group for walking and biking.

Characteristics

651

(n)

100

(%)

Characteristics

660

(%)

100

(%)

Gender

Participated in types of walking/biking

Male

321

49.3

Leisure-related activities

216

33.2

Female

330

50.7

Tourism-related activities

217

33.3

Other

0

0.0

Work-related activities

218

33.5

Age

Reason for walking/biking

Between 18 and 29 years old

117

18.0

Self-satisfaction

268

41.2

Between 30 and 39 years old

115

17.7

Experiencing the community

77

11.8

Between 40 and 49 years old

144

22.1

Mental wellbeing and health

236

36.3

Between 50 and 59 years old

149

22.8

Physical wellbeing and health

436

67.0

60 years old and over

126

19.4

Opportunity to socialize

82

12.6

Educational level

Contact with nature

267

41.0

Less than or high school diploma

117

18.0

Visiting attractions

149

22.9

2-year college

99

15.2

Protecting the environment

91

14.0

University

357

54.8

Access to public transport

177

27.2

Graduate school or higher

78

12.0

Access to shopping

129

19.8

Marital status

Walking/biking with a dog

37

5.7

Single

240

36.9

Opportunity to be alone

118

18.1

Married

403

61.9

Opportunity to be with family

112

17.2

Other

8

1.2

Other

26

4.0

Monthly household income

More walking/biking compared to before the COVID-19

Less than KRW 2.00-3.99 million

264

40.6

Yes

155

23.8

From KRW 4.00 to 7.99 million

294

45.1

No

293

45.0

KRW 8.00 million or more

93

14.3

Same

203

31.2

Occupation

Used smart applications for walking/biking

Professional (e.g., attorney, engineer)

66

10.1

Yes

421

64.7

Business owner/self-employed

44

6.8

No

230

35.3

Service worker

73

11.2

Used types of smart applications for walking/biking

Office/administrative/clerical worker

235

36.0

GPS/Maps (e.g., tracker, route)

242

37.2

Civil servant (government)

29

4.5

Fitness (e.g., calorie counting)

173

26.6

Home maker

76

11.7

Counter (e.g., step or distance measurement)

285

43.8

Student

35

5.4

Heart rate (e.g., pulse measurement)

103

15.8

Retiree

21

3.2

Safety (e.g., CCTV location)

23

3.5

Unemployed

29

4.5

Amenity (e.g., toilet, shelter, facilities)

43

6.6

Other

43

6.6

Augmented reality apps

11

1.7

Residential area

Other

16

2.5

Seoul-metropolitan area

428

65.6

Worry about personal safety when walking/biking

Non-metropolitan area

223

34.4

Disagree

253

38.8

Duration of answering the survey

Neither agree nor disagree

137

21.0

Between 5 and 533.8 minutes

651

100.0

Agree

261

40.2

Riding e-bike

Companions when walking/biking

Yes, I mostly ride electric bikes

27

4.1

Alone

386

59.3

No, I mostly ride conventional bikes

624

95.9

Friends

82

12.6

Providing thoughtful/honest answers

Family/Relatives

162

24.9

Yes

651

100.0

Coworkers

19

2.9

No

0

0.0

Other

2

0.3

 Note: The walking activity group has 325 cases, while the biking activity group has 326 cases.

Appendix D. Mediating (indirect) effects.

Path

Direct effect

Indirect effect

Total effect

t-vale

p-value

f2

Awareness of public green space → Attitude

0.163***

0.163***

3.708

<0.001

0.029

Awareness of public green space → Desire

-0.002ns

0.006ns

0.004ns

0.733

>0.05

Awareness of public green space → Behavioral intention

0.159***

0.001ns

0.160***

4.734

<0.001

0.043

Motivation → Attitude

0.539***

0.539***

12.805

<0.001

0.319

Motivation → Desire

-0.083ns

0.020ns

-0.060ns

0.795

>0.05

0.005

Motivation → Behavioral intention

0.535***

0.022*

0.557***

16.958

<0.001

0.390

Attitude → Desire

0.037ns

0.037ns

0.857

>0.05

0.002

Attitude → Behavioral intention

0.008ns

0.008ns

0.824

>0.05

Subjective norm → Desire

0.137***

0.137***

3.473

<0.001

0.027

Subjective norm → Behavioral intention

0.030***

0.030***

3.225

<0.001

Perceived behavioral control → Desire

0.003ns

0.003ns

0.080

>0.05

0.000

Perceived behavioral control → Behavioral intention

0.001ns

0.001ns

0.081

>0.05

Positive anticipated emotion → Desire

0.466***

0.466***

10.048

<0.001

0.204

Positive anticipated emotion → Behavioral intention

0.102***

0.102***

5.775

<0.001

Negative anticipated emotion → Desire

0.225***

0.225***

5.997

<0.001

0.087

Negative anticipated emotion → Behavioral intention

0.049***

0.049***

4.731

<0.001

Desire → Behavioral intention

0.219***

0.227***

7.127

<0.001

0.082

Note: ***p<0.001; ns = non-significant.

Appendix E. Grouping the moderator.

Construct

Cronbach alpha

Group

Sample size

Mean

Perceived usefulness of smart applications

0.817

High

433

5.36

Low

218

3.61

There are a lot of references, which is very good, but it should be revised (e.g. reference should not appear in the conclusion chapter).

RESPONSE 2: Thank you so much for bringing the issue. As you suggested, we have deleted four references in the conclusion as follows (please also see the conclusion of the revised manuscript):

Gössling, S. ICT and Transport Behavior: A Conceptual Review. Int J Sustain Transp 2018, 12 (3), 153–164. https://doi.org/10.1080/15568318.2017.1338318.

Hall, C. M.; Le-Klähn, D. T.; Ram, Y. Tourism, Public Transport and Sustainable Mobility; Channel View Publications, 2017.

Hall, C. M.; Ram, Y.; Shoval, N. The Routledge International Handbook of Walking, 1st ed.; Routledge: London, 2018. https://doi.org/10.4324/9781315638461.

Rasoolimanesh, S. M.; Seyfi, S.; Rather, R. A.; Hall, C. M. Investigating the Mediating Role of Visitor Satisfaction in the Relationship between Memorable Tourism Experiences and Behavioral Intentions in Heritage Tourism Context. Tour Rev 2022, 77 (2), 687–709. https://doi.org/10.1108/TR-02-2021-0086.

 Thank you very much for your constructive comments!

Reviewer 2 Report

Title

Overall, the title of the manuscript reads well and is an accurate presentation of the manuscript content.

Abstract

Overall, the abstract is well written, I have only some concerns.

The term “public green spaces” might not be understandable for the reader.

Methods part is missing from the abstract.

Results in abstract would benefit from accurate values.

Introduction and literature review – it is hard to understand why Authors choose to distinguish between introduction and literature review sections, it is not common in academic papers. I recommend combining those sections. Also, objective of the study (from introduction) could be moved to the hypothesis development section.

Motivation theory (2.1.2) – the motivation theory sections needs more in depth overview. Please elaborate this section, Authors are recommended to study work by Deci and Ryan and elaborate on intrinsic and extrinsic forms of motivation. Also, it is important to note that it is specifically intrinsic motivation that is found to be related with objectively measured physical activity (Kalajas-Tilga et al., 2020).

Kalajas-Tilga, H., Koka, A., Hein, V., Tilga, H., & Raudsepp, L. (2020). Motivational processes in physical education and objectively measured physical activity among adolescents. Journal of Sport and Health Science, 9(5), 462–471. https://doi.org/10.1016/j.jshs.2019.06.001

Hypothesis development – Figure 1 could be more readable.

Methods

Methods section is clear and very detailed.

Results

Please check Table 1, there are some visual errors.

Table 2, Table 3 and Table 4 – headings of those tables could be more specific.

Figure 2 – the figure could be more readable.

Table 4 – there are some typos, please check “t-vale”.

Table 5 – the heading is clearly too vague.

Table 6 – please fit this table on one page, currently it seems quite odd.

Discussion section is clearly too short, less than one page. Authors have several results that could be discussed. Authors are strongly recommended to elaborate this section.

Limitations and future research directions – please also add strengths of the currents study.

Author Response

Reviewer 2:

English language and style

( ) Extensive editing of English language and style required

( ) Moderate English changes required

(x) English language and style are fine/minor spell check required

( ) I don't feel qualified to judge about the English language and style

Thank you so much for your thoughtful comment. As you suggested, we have conducted extensive editing of English language and style by professional editors and authors. Please refer to the blue text of the entire revised manuscript.

Yes      Can be improved        Must be improved       Not applicable

Does the introduction provide sufficient background and include all relevant references?

( )        ( )        (x)       ( )

Are all the cited references relevant to the research?

( )        (x)       ( )        ( )

Is the research design appropriate?

( )        (x)       ( )        ( )

Are the methods adequately described?

( )        (x)       ( )        ( )

Are the results clearly presented?

( )        (x)       ( )        ( )

Are the conclusions supported by the results?

( )        (x)       ( )        ( )

Comments and Suggestions for Authors

Title

Overall, the title of the manuscript reads well and is an accurate presentation of the manuscript content.

We appreciate your valuable feedback and comments.

Abstract

Overall, the abstract is well written, I have only some concerns.

The term “public green spaces” might not be understandable for the reader.

Methods part is missing from the abstract.

Results in abstract would benefit from accurate values.

RESPONSE 1: Thank you so much for your insightful comments. As you suggested, we have revised the abstract, adding public green spaces’ examples, the methods part, and provided accurate values of the results as follows (please also refer to lines 16-33 in the blue text):

Abstract: Public green spaces (e.g., parks, green trails, greenways) and motivations to engage in active transport are essential for encouraging walking and biking. However, how these key factors influence walker and biker behavior is potentially being increasingly influenced by the use of smart apps as they become more ubiquitous in everyday practices. To fill this research gap, this work creates and tests a theoretically integrated study framework grounded on an extended model of goal-directed behavior including public green space and motivation with perceived usefulness of smart apps. In order to accomplishment the purpose of this study, we conducted online survey to Korean walkers (n=325) and bikers (n=326) during July 10-25, 2021, and applied partial least squares, structural equation, and multi-group analysis to validate the research model. Results revealed that active transport users’ awareness of public green space positively influences attitude toward (γ = 0.163), as well as behavioral intention for (γ = 0.159), walking and biking. Also, motivation (extrinsic and intrinsic) greatly influences attitude (γ = 0.539) and behavioral intention (γ = 0.535). Subjective norms (γ = 0.137) and positive (γ = 0.466) and negative anticipated emotions (γ = 0.225) have a significant impact on desire that leads to behavioral intention. High and low perceived smart app usefulness also significantly moderate between public green space and attitude (t-value = 25.705), public green space and behavioral intention (t-value = 25.726), motivation and attitude (t-value = -25.561), and motivation and behavioral intention (t-value = -15.812). Consequently, the findings contribute to academics and practitioners by providing new knowledge and insights.

Introduction and literature review – it is hard to understand why Authors choose to distinguish between introduction and literature review sections, it is not common in academic papers. I recommend combining those sections.

RESPONSE 2: Thank you for raising the issue. As you suggested, we have combined the introduction and literature review sections. Please see the blue text between lines 54 – 248 of our revised manuscript.

Also, objective of the study (from introduction) could be moved to the hypothesis development section.

RESPONSE 3: With regard to your valuable suggestion, we have moved the objective of the study from the introduction to the hypothesis development as follows (please also see lines 141- 151):

2.1. Hypothesis development

The object of the work is therefore to create and assess a conceptual framework based on an extended MGB (EMGB), adding key variables of awareness of public green space and extrinsic and intrinsic motivation, along with moderating effect of perceived usefulness of smart apps. This study raises three research questions: Do awareness of public green space and motivation for using active transport influence attitude, desire, and behavioral intention to walk and bike?; does an EMGB model predict walkers and bikers’ behavioral intention?; and does the perceived usefulness of smart apps moderate among awareness, motivation, attitude, desire, and behavior? Given the importance of active transport and greenspace for public and environmental health the findings of the work potentially offer valuable academic and managerial insights for research and ac-tion on active transport initiatives.

Motivation theory (2.1.2) – the motivation theory sections needs more in depth overview. Please elaborate this section, Authors are recommended to study work by Deci and Ryan and elaborate on intrinsic and extrinsic forms of motivation. Also, it is important to note that it is specifically intrinsic motivation that is found to be related with objectively measured physical activity (Kalajas-Tilga et al., 2020).

RESPONSE 4: Based on your thoughtful suggestion, we have added one paragraph regarding intrinsic and extrinsic forms of motivation by Deci and Ryan as well as Kalajas-Tilga et al. as follows (please refer to lines 81 - 85):

Specifically, intrinsic motivation remains an important construct, reflecting the natural human propensity to learn and assimilate; however, extrinsic motivation may vary con-siderably in its relative autonomy and thus can either reflect external control or true self-regulation [31-32]. Intrinsic motivation is also found to be positively related to moderate-to-vigorous physical activity [33].

  1. Deci, E. L.; Koestner, R.; Ryan, R. M. 30. A Meta-Analytic Review of Experiments Examining the Effects of Extrinsic Rewards on Intrinsic Motivation. Psychol Bull 1999, 125 (6), 627–668. https://doi.org/10.1037/0033-2909.125.6.627.
  2. Ryan, R. M.; Deci, E. L. 30. Intrinsic and Extrinsic Motivations: Classic Definitions and New Directions. Contemp Educ Psychol 2000, 25 (1), 54–67. https://doi.org/10.1006/ceps.1999.1020.
  3. Kalajas-Tilga, H.; Koka, A.; Hein, V.; Tilga, H.; Raudsepp, L. Motivational Processes in Physical Education and Objectively Measured Physical Activity among Adolescents. J Sport Heal Sci 2020, 9 (5), 462–471. https://doi.org/10.1016/j.jshs.2019.06.001.

Kalajas-Tilga, H., Koka, A., Hein, V., Tilga, H., & Raudsepp, L. (2020). Motivational processes in physical education and objectively measured physical activity among adolescents. Journal of Sport and Health Science, 9(5), 462–471. https://doi.org/10.1016/j.jshs.2019.06.001

RESPONSE 5: Thank you for your kind recommendation of the useful article. We have carefully read and cited the reference above.

Hypothesis development – Figure 1 could be more readable.

RESPONSE 6: In line with your valuable suggestion, we have improved Figure 1 to be more readable as follows (please see lines 245-246 of our revised manuscript):

Fig. 1. Proposed research model.

Methods

Methods section is clear and very detailed.

RESPONSE 7: Thank you very much for your favorable feedback!

Results

Please check Table 1, there are some visual errors.

RESPONSE 8: Thank you for raising the issue. As you suggested, we have revised some visual errors as follows (please also refer to lines 817-818):

Appendix C. Demographic characteristic and general information of the entire group for walking and biking.

Characteristics

651

(n)

100

(%)

Characteristics

651

(n)

100

(%)

Gender

Participated in types of walking/biking

Male

321

49.3

Leisure-related activities

216

33.2

Female

330

50.7

Tourism-related activities

217

33.3

Other

0

0.0

Work-related activities

218

33.5

Age

Reason for walking/biking

Between 18 and 29 years old

117

18.0

Self-satisfaction

268

41.2

Between 30 and 39 years old

115

17.7

Experiencing the community

77

11.8

Between 40 and 49 years old

144

22.1

Mental wellbeing and health

236

36.3

Between 50 and 59 years old

149

22.8

Physical wellbeing and health

436

67.0

60 years old and over

126

19.4

Opportunity to socialize

82

12.6

Educational level

Contact with nature

267

41.0

Less than or high school diploma

117

18.0

Visiting attractions

149

22.9

2-year college

99

15.2

Protecting the environment

91

14.0

University

357

54.8

Access to public transport

177

27.2

Graduate school or higher

78

12.0

Access to shopping

129

19.8

Marital status

Walking/biking with a dog

37

5.7

Single

240

36.9

Opportunity to be alone

118

18.1

Married

403

61.9

Opportunity to be with family

112

17.2

Other

8

1.2

Other

26

4.0

Monthly household income

More walking/biking compared to before the COVID-19

Less than KRW 2.00-3.99 million

264

40.6

Yes

155

23.8

From KRW 4.00 to 7.99 million

294

45.1

No

293

45.0

KRW 8.00 million or more

93

14.3

Same

203

31.2

Occupation

Used smart applications for walking/biking

Professional (e.g., attorney, engineer)

66

10.1

Yes

421

64.7

Business owner/self-employed

44

6.8

No

230

35.3

Service worker

73

11.2

Used types of smart applications for walking/biking

Office/administrative/clerical worker

235

36.0

GPS/Maps (e.g., tracker, route)

242

37.2

Civil servant (government)

29

4.5

Fitness (e.g., calorie counting)

173

26.6

Home maker

76

11.7

Counter (e.g., step or distance measurement)

285

43.8

Student

35

5.4

Heart rate (e.g., pulse measurement)

103

15.8

Retiree

21

3.2

Safety (e.g., CCTV location)

23

3.5

Unemployed

29

4.5

Amenity (e.g., toilet, shelter, facilities)

43

6.6

Other

43

6.6

Augmented reality apps

11

1.7

Residential area

Other

16

2.5

Seoul-metropolitan area

428

65.6

Worry about personal safety when walking/biking

Non-metropolitan area

223

34.4

Disagree

253

38.8

Duration of answering the survey

Neither agree nor disagree

137

21.0

Between 5 and 533.8 minutes

651

100.0

Agree

261

40.2

Riding e-bike

Companions when walking/biking

Yes, I mostly ride electric bikes

27

4.1

Alone

386

59.3

No, I mostly ride conventional bikes

624

95.9

Friends

82

12.6

Providing thoughtful/honest answers

Family/relatives

162

24.9

Yes

651

100.0

Coworkers

19

2.9

No

0

0.0

Other

2

0.3

 Note: The walking activity group has 325 cases, while the biking activity group has 326 cases.

Table 2, Table 3 and Table 4 – headings of those tables could be more specific.

RESPONSE 9: Thank you for your keen observation on our manuscript. As you suggested, we have specified the heading of Table 2 and Appendix D which are originally Tables 3 and 4 as follows (see lines 397-398 and 820-822): 

Table 2. CFA on the measurements, descriptive statistics, and normal distribution.

Constructs

Factor

loading

t-

value

Me-an

SD*

Kurto-

sis

Skew-

ness

VIF**

Awareness of public green space

1. I am interested in public green space for walking/biking.

0.864

71.650

5.261

1.218

0.394

-0.653

2.685

2. I am aware of public green spaces for walking/biking.

0.693

24.276

4.939

1.343

0.173

-0.628

1.486

3. I care about public green trails for walking/biking.

0.866

75.457

5.427

1.164

0.640

-0.682

2.701

4. Public green spaces provide cool areas in which to walk/bike when it is hot.

0.772

37.634

5.498

1.135

0.496

-0.727

1.782

5. Public green spaces are attractive to walk/bike any time of year.

0.787

35.861

5.450

1.154

0.503

-0.692

1.846

Extrinsic motivation

1. Walking/biking improves my personal health.

0.729

31.041

5.937

0.949

1.597

-0.987

1.562

2. Walking/biking contributes to the environment.

0.903

70.813

5.533

1.120

0.460

-0.665

3.565

3. Walking/biking contributes to mitigating climate change.

0.891

60.295

5.396

1.182

1.003

-0.789

3.558

4. Walking/biking contributes to lowering air pollution

0.895

94.862

5.512

1.212

0.896

-0.849

3.520

5. Walking/biking improves public health.

0.802

42.136

5.144

1.118

0.120

-0.333

1.999

Intrinsic motivation

1. Walking/biking is enjoyable for me.

0.911

112.620

5.287

1.164

0.363

-0.551

3.366

2. Walking/biking brings me self-satisfaction.

0.896

95.143

5.298

1.168

0.434

-0.550

2.887

3. Walking/biking makes me happy.

0.915

132.335

5.169

1.178

0.199

-0.387

3.484

4. I walk for refreshment.

0.824

38.645

5.339

1.163

1.025

-0.763

2.037

Attitude to active transport

1. Walking/biking is an affirmative behavior.

0.881

90.493

5.690

1.009

1.041

-0.759

2.563

2. Walking/biking is a beneficial behavior.

0.864

67.478

5.730

0.987

0.524

-0.648

2.418

3. Walking/biking is an essential behavior.

0.713

26.581

4.763

1.428

-0.231

-0.397

1.498

4. Walking/biking is a legitimate behavior.

0.857

67.214

5.255

1.100

0.275

-0.406

2.213

Subjective norm on active transport

1. Most people who are important to me think I should walk/bike.

0.860

65.546

4.293

1.409

-0.391

-0.223

2.445

2. Most people who are important to me would want me to walk/bike.

0.897

82.026

4.516

1.331

-0.028

-0.288

3.016

3. Most people who are important to me support my walking/biking.

0.819

37.580

4.980

1.244

0.633

-0.630

2.010

4. Most people who are important to me take pride that I go walking/biking.

0.869

63.156

4.429

1.324

0.314

-0.345

2.187

Perceived behavioral control

1. Walking/biking or not is entirely up to me.

0.652

16.000

5.954

0.997

0.931

-0.990

1.412

2. I can walk/bike whenever I want.

0.786

29.999

5.320

1.317

0.266

-0.798

1.683

3. I have the physical strength to walk/bike.

0.750

23.633

5.696

1.025

0.139

-0.686

1.365

4. I have time to walk/bike.

0.848

49.348

5.281

1.137

0.268

-0.542

1.696

Positive anticipated emotion

1. If I walk/bike, I will feel excited.

0.900

85.682

5.157

1.203

0.393

-0.521

3.098

2. If I walk/bike, I will feel glad.

0.920

117.165

5.210

1.152

0.728

-0.556

3.782

3. If I walk/bike, I will feel satisfied.

0.887

88.516

5.461

1.031

0.669

-0.582

2.747

4. If I walk/bike, I will feel happy.

0.915

124.078

5.252

1.140

0.349

-0.449

3.576

Negative anticipated emotion

1. If I can't walk/bike, I will be angry.

0.921

120.119

3.989

1.736

-0.848

-0.035

3.881

2. If I can't walk/bike, I will be disappointed.

0.932

157.084

4.607

1.695

-0.762

-0.390

4.200

3. If I can't walk/bike, I will be worried.

0.902

76.368

4.539

1.687

-0.869

-0.243

3.285

4. If I can't walk/bike, I will be sad.

0.937

174.769

4.458

1.750

-0.848

-0.278

4.488

Desire on walking/biking

1. I do want to walk/bike.

0.847

67.976

5.167

1.197

0.712

-0.670

2.122

2. My desire to walk/bike is passionate.

0.914

130.799

4.525

1.367

0.022

-0.431

3.495

3. I am enthusiastic to walk/bike.

0.886

70.491

4.038

1.382

-0.267

-0.146

3.151

4. I am eager to walk/bike.

0.916

128.309

4.258

1.397

-0.201

-0.308

3.827

Behavioral intention on active transport

1. To increase my personal well-being, I'm planning to walk/bike.

0.785

42.778

5.301

1.099

0.746

-0.652

1.951

2. To improve my personal health, I will make an effort to walk/bike.

0.740

26.692

5.605

1.093

1.104

-0.861

1.771

3. To mitigate climate change, I am willing to walk/bike.

0.822

54.767

4.900

1.275

0.310

-0.588

2.574

4. To protect the environment, I do intend to walk/bike.

0.822

50.032

4.693

1.372

-0.082

-0.387

2.795

5. To increase public well-being, I'm planning to walk/bike.

0.792

42.040

4.584

1.312

-0.037

-0.259

1.973

Perceived usefulness of smart applications

1. I believe that using smart applications for walk/bike would enable me to accomplish walking/biking better.

0.897

89.006

4.628

1.244

0.067

-0.266

2.880

2. I believe that using smart applications for walk/bike would improve my walking/biking performance.

0.885

73.279

4.866

1.22

0.197

-0.358

2.871

3. I believe that using smart applications for walk/bike would make it easier to do my walking/biking.

0.861

60.945

4.567

1.329

-0.124

-0.304

2.238

4. I believe that using smart applications for walk/bike would enhance my effectiveness on walking/biking.

0.876

72.385

5.022

1.149

0.724

-0.493

2.664

Note: The italics indicates non-normal distribution. *Standard deviation. **Variance inflation factor

Appendix D. Mediating (indirect) effects on the proposed research model.

Path

Direct effect

Indirect effect

Total effect

t-vale

p-value

f2

Awareness of public green space → Attitude

0.163***

0.163***

3.708

<0.001

0.029

Awareness of public green space → Desire

-0.002ns

0.006ns

0.004ns

0.733

>0.05

Awareness of public green space → Behavioral intention

0.159***

0.001ns

0.160***

4.734

<0.001

0.043

Motivation → Attitude

0.539***

0.539***

12.805

<0.001

0.319

Motivation → Desire

-0.083ns

0.020ns

-0.060ns

0.795

>0.05

0.005

Motivation → Behavioral intention

0.535***

0.022*

0.557***

16.958

<0.001

0.390

Attitude → Desire

0.037ns

0.037ns

0.857

>0.05

0.002

Attitude → Behavioral intention

0.008ns

0.008ns

0.824

>0.05

Subjective norm → Desire

0.137***

0.137***

3.473

<0.001

0.027

Subjective norm → Behavioral intention

0.030***

0.030***

3.225

<0.001

Perceived behavioral control → Desire

0.003ns

0.003ns

0.080

>0.05

0.000

Perceived behavioral control → Behavioral intention

0.001ns

0.001ns

0.081

>0.05

Positive anticipated emotion → Desire

0.466***

0.466***

10.048

<0.001

0.204

Positive anticipated emotion → Behavioral intention

0.102***

0.102***

5.775

<0.001

Negative anticipated emotion → Desire

0.225***

0.225***

5.997

<0.001

0.087

Negative anticipated emotion → Behavioral intention

0.049***

0.049***

4.731

<0.001

Desire → Behavioral intention

0.219***

0.227***

7.127

<0.001

0.082

Note: ***p<0.001; ns = non-significant.

Figure 2 – the figure could be more readable.

RESPONSE 10: In compliance with your suggestion, we have improved the readability of the Figure 2 as follows (see lines 430-431):

Fig. 2. Results of path analysis.

Table 4 – there are some typos, please check “t-vale”.

RESPONSE 11: According to your suggestion, we have checked some typos in Appendix D of the original Table 4 as follows (also refer to lines 820-822):

Appendix D. Mediating (indirect) effects on the proposed research model.

Path

Direct effect

Indirect effect

Total effect

t-value

p-value

f2

Awareness of public green space → Attitude

0.163***

0.163***

3.708

<0.001

0.029

Awareness of public green space → Desire

-0.002ns

0.006ns

0.004ns

0.733

>0.05

Awareness of public green space → Behavioral intention

0.159***

0.001ns

0.160***

4.734

<0.001

0.043

Motivation → Attitude

0.539***

0.539***

12.805

<0.001

0.319

Motivation → Desire

-0.083ns

0.020ns

-0.060ns

0.795

>0.05

0.005

Motivation → Behavioral intention

0.535***

0.022*

0.557***

16.958

<0.001

0.390

Attitude → Desire

0.037ns

0.037ns

0.857

>0.05

0.002

Attitude → Behavioral intention

0.008ns

0.008ns

0.824

>0.05

Subjective norm → Desire

0.137***

0.137***

3.473

<0.001

0.027

Subjective norm → Behavioral intention

0.030***

0.030***

3.225

<0.001

Perceived behavioral control → Desire

0.003ns

0.003ns

0.080

>0.05

0.000

Perceived behavioral control → Behavioral intention

0.001ns

0.001ns

0.081

>0.05

Positive anticipated emotion → Desire

0.466***

0.466***

10.048

<0.001

0.204

Positive anticipated emotion → Behavioral intention

0.102***

0.102***

5.775

<0.001

Negative anticipated emotion → Desire

0.225***

0.225***

5.997

<0.001

0.087

Negative anticipated emotion → Behavioral intention

0.049***

0.049***

4.731

<0.001

Desire → Behavioral intention

0.219***

0.227***

7.127

<0.001

0.082

Note: ***p<0.001; ns = non-significant.

Table 5 – the heading is clearly too vague.

RESPONSE 12: Thank you for bringing the issue. As you suggested, we have cleared the heading of Appendix E (the original Table 5) as follows (also see lines 824-825):

               Appendix E. Grouping the moderator of smart app usefulness.

Construct

Cronbach alpha

Group

Sample size

Mean

Perceived usefulness of smart applications

0.817

High

433

5.36

Low

218

3.61

Table 6 – please fit this table on one page, currently it seems quite odd.

RESPONSE 13: Thank you for raising the issue. As you suggested, we have revised Table 3 (previously Table 6) as follows (see the blue text in lines 455-456 of our revised manuscript):

Table 3. Moderating role of high and low perceived usefulness of smart apps.

H5

Path

High group

(A)

Low

Group

(B)

A-B

t-

value

p value

Hypothesis

test

H5a

Awareness of public green space → Attitude

0.206***

0.076ns

0.130

25.705

<0.001

Supported

H5b

Awareness of public green space → Desire

0.017ns

-0.068ns

0.085

20.052

ns

Not supported

H5c

Awareness of public green space → Behavioral intention

0.202***

0.099ns

0.102

25.726

<0.001

Supported

H5d

Motivation for walking/biking → Attitude

0.470***

0.585***

-0.115

-25.561

<0.001

Supported

H5e

Motivation for walking/biking → Desire

0.086ns

0.027ns

0.058

10.128

ns

Not supported

H5f

Motivation for walking/biking → Behavioral intention

0.494***

0.561***

-0.067

-15.812

<0.001

Supported

***p<0.001; ns = non-significant. Since the two hypotheses in the high and low groups are insignificant, H5b and H5e are not supported.

Discussion section is clearly too short, less than one page. Authors have several results that could be discussed. Authors are strongly recommended to elaborate this section.

RESPONSE 14: Thank you so much for your insightful comment. As you suggested, we have further elaborated the discussion section as follows (please also refer to lines 459-500):

  1. Discussion

This work suggests that people with higher awareness of public green space (e.g., parks, gardens, forests, greenbelts, greenways, institutional green spaces) are more likely to have positive attitude toward, as well as behavioral intention for participation in, walking and biking. Although such results may seem readily apparent, they actually extend previous literature on personal values and public green space in active transport in relation to attitude and behavior in urban areas [47]. The strong relationships between motivation and attitude as well as motivation and behavioral intention suggest that individuals with strong motivation for walking and biking are more likely to have better attitude toward as well as behavioral intention for participation in active transport behavior. These findings reinforce previous research on individuals’ motivations and satisfaction with respect to walking trips [35] and, perhaps more significantly, provide insights into more effective behavioral interventions that seek to encourage walking and cycling. This is particularly important as extrinsic motivations are found to be more significant than intrinsic ones in terms of wanting to walk and bike, extending previous findings on intrinsic and extrinsic motivations applied by self-determination theory [31-33]. 

The positive influences of subjective norms, positive anticipated emotion, and negative anticipated emotion on desire for walking and biking are substantially significant. This work therefore expands previous findings on MGB in the context of active travel behavior for choice cycle [19] and sustainable transport for cities [20]. In particular, the insignificant relationships between awareness and desire, motivation and desire, perceived behavioral control and desire, and attitude and desire suggest that people’s desires to walk and bike are not influenced by awareness, motivation, perceived behavioral control, and attitude. A possible reason for these findings is that walking and biking are essential to Korean residents who are dealing with the COVID-19 pandemic so that even without the influences of awareness, motivation, attitude, and perceived control on desire, Koreans still continue to walk and bike for leisure, tourism, and/or work. These findings are partially consistent with findings of prior research on support for the sustainable development goals (SDGs) in Korea [17] 

With regard to the moderating effect of smart app usefulness, the stronger relationships between awareness of public green space and attitude and between awareness of public green space and behavioral intention from the high perceived group of smart app usefulness than the low group implies that a person with higher awareness of public green space for active transport tend to have stronger attitudes and behavioral intentions to walking and biking if the person has higher perceived usefulness of smart apps. These findings extend the literature on using apps for active transport in the context of smart cities [21-22]. On the other hand, the low perceived group of smart app usefulness has greater impacts on the relationships between motivation and attitude as well as motivation and behavioral intention compared to the high perceived group on smart app usefulness. These results are unexpected and contrary on our hypotheses and previous studies on fuzzy logic app for pedestrians [21]

Limitations and future research directions – please also add strengths of the currents study.

RESPONSE 15: We appreciate your raising the issue. In order to address your concern, we have added some strengths of the currents study as follows (please also see lines 539-557): 

  1. Limitations and Future Research Directions

Although this work makes a number of contributions, and has theoretical and practical implications, it also has some limitations that provide future study opportunities. First, this survey was only implemented in one country (South Korea) and during the COVID-19 pandemic (from July 10 to July 25, 2021) so generalization of the findings needs to be carefully considered, with further research in other cultures and active transport environments required. Second, although this study outlines the importance of understanding walking and biking behaviors in the specific context they occur, e.g. tourism, leisure, and work, more detailed examination of the different types of walking and cycling activities would be extremely valuable with respect to the purpose of such activities and their connection to other transport modes. Third, even though this study has applied PLS-SEM and MGA, the adoption of other research methods such as Fuzzy-set Qualitative Comparative Analysis (fsQCA) and in-depth interviews would also be valuable to more deeply understand the sequential relationships existing among stated attitudes, behavioral intentions, and actual behaviors, particularly with smart app use. Furthermore, although this study sheds light on walking and biking behavior in terms of green space, motivation, and MGB, future study on the differences among tourism, leisure, and work activities would be valuable to better understand walking and biking behavior.

Thank you very much for your valuable review on our research!

Reviewer 3 Report

The presented manuscript applies an extended model of goal-directed behaviour to study if the presence of public green space increases intention and or motivation to use active transport modes, such as walking or cycling. While the methodology is sound and clearly presented the argumentation for the need of this research needs to be improved.  The title suggests the main focus of this research lies on impact of smartphone apps in regard to active transport behaviour, however, this is just a tiny section of this research. Thus, I suggest the authors change the title of their manuscript to something less misleading, e.g. “Application of EMGB to study impacts of public green space on active transport behaviour. Evidence from South Korea.”

Secondly, the question of how the street environment affects active transport behaviour is well covered within the built environment literature, thus results aren’t contributing much to the existing body of literature. It would also be good to provide a definition of the understanding of “public green space” within the scope of this paper.

Thirdly, the authors should better explain why they distinguish between daily mobility, leisure and tourism in their questionnaire, although they do not present results for these different motivations?

Lastly some minor editorial comments:

·        Written expression and correctness need to be improved from section 3 onwards

·        Table 2, 5th column header: Why n = 660? (Also there is % instead of n)

·        Section 5 Conclusion should be section 6

Author Response

Reviewer 3:

English language and style

( ) Extensive editing of English language and style required

(x) Moderate English changes required

( ) English language and style are fine/minor spell check required

( ) I don't feel qualified to judge about the English language and style

Thank you so much for your thoughtful comment. As you suggested, we have conducted extensive editing of English language and style by professional editors and authors. Please refer to the blue text of the entire revised manuscript.

Yes      Can be improved        Must be improved       Not applicable

Does the introduction provide sufficient background and include all relevant references?

( )        (x)       ( )        ( )

Are all the cited references relevant to the research?

(x)       ( )        ( )        ( )

Is the research design appropriate?

(x)       ( )        ( )        ( )

Are the methods adequately described?

(x)       ( )        ( )        ( )

Are the results clearly presented?

(x)       ( )        ( )        ( )

Are the conclusions supported by the results?

( )        ( )        (x)       ( )

Comments and Suggestions for Authors

The presented manuscript applies an extended model of goal-directed behaviour to study if the presence of public green space increases intention and or motivation to use active transport modes, such as walking or cycling. While the methodology is sound and clearly presented the argumentation for the need of this research needs to be improved.

We appreciate your valuable feedback and comments.

The title suggests the main focus of this research lies on impact of smartphone apps in regard to active transport behaviour, however, this is just a tiny section of this research. Thus, I suggest the authors change the title of their manuscript to something less misleading, e.g. “Application of EMGB to study impacts of public green space on active transport behaviour. Evidence from South Korea.”

RESPONSE 1: Thank you so much for your valuable comment. As you suggested, we have changed the title as follows (also refer to lines 2-3 in the blue text):

New title: Application of EMGB to study impacts of public green space on active transport behavior: Evidence from South Korea

Secondly, the question of how the street environment affects active transport behaviour is well covered within the built environment literature, thus results aren’t contributing much to the existing body of literature. It would also be good to provide a definition of the understanding of “public green space” within the scope of this paper.

RESPONSE 2: We appreciate your valuable comment. As you suggested, we have added the definition of the public green space in this study as follows (also see lines 69-74):

Therefore, this study considers public green space as an important public good that can encourage active transport of walking and biking and contribute to more sustainable urban environments. Thus, public green space is defined as parks, gardens, forests, woods, greenbelts, greenways and green trails, such green space can improve mental and physical health, enable personal and public well-being, and contribute to environmental and/or climate change mitigation goals [4-10].

Thirdly, the authors should better explain why they distinguish between daily mobility, leisure and tourism in their questionnaire, although they do not present results for these different motivations?

RESPONSE 3: Thank you very much for bringing the issue. In order to resolve your concern, we have provided the future research direction since we have missed to study on the differences of mobility, leisure, and tourism as follows (refer to lines 553-557):

Furthermore, although this study sheds light on walking and biking behavior in terms of green space, motivation, and MGB, future study on the differences among tourism, lei-sure, and work activities would be valuable to better understand walking and biking behavior..

Lastly some minor editorial comments:

  • Written expression and correctness need to be improved from section 3 onwards

RESPONSE 4: Thank you for bringing the issue. In order to address your concern, we have thoroughly review and edit the whole papers, checking the written expressions and correctnesses. Please refer to the blue text of our revised manuscript.

  • Table 2, 5th column header: Why n = 660? (Also there is % instead of n)

RESPONSE 5: Thank you very much for your keen observation on our manuscript. As you commented, we have revised the typos in Appendix C as follows (also see lines 817-818):

Appendix C. Demographic characteristic and general information of the entire group for walking and biking.

Characteristics

651

(n)

100

(%)

Characteristics

651

(n)

100

(%)

Gender

Participated in types of walking/biking

Male

321

49.3

Leisure-related activities

216

33.2

Female

330

50.7

Tourism-related activities

217

33.3

Other

0

0.0

Work-related activities

218

33.5

Age

Reason for walking/biking

Between 18 and 29 years old

117

18.0

Self-satisfaction

268

41.2

Between 30 and 39 years old

115

17.7

Experiencing the community

77

11.8

Between 40 and 49 years old

144

22.1

Mental wellbeing and health

236

36.3

Between 50 and 59 years old

149

22.8

Physical wellbeing and health

436

67.0

60 years old and over

126

19.4

Opportunity to socialize

82

12.6

Educational level

Contact with nature

267

41.0

Less than or high school diploma

117

18.0

Visiting attractions

149

22.9

2-year college

99

15.2

Protecting the environment

91

14.0

University

357

54.8

Access to public transport

177

27.2

Graduate school or higher

78

12.0

Access to shopping

129

19.8

Marital status

Walking/biking with a dog

37

5.7

Single

240

36.9

Opportunity to be alone

118

18.1

Married

403

61.9

Opportunity to be with family

112

17.2

Other

8

1.2

Other

26

4.0

Monthly household income

More walking/biking compared to before the COVID-19

Less than KRW 2.00-3.99 million

264

40.6

Yes

155

23.8

From KRW 4.00 to 7.99 million

294

45.1

No

293

45.0

KRW 8.00 million or more

93

14.3

Same

203

31.2

Occupation

Used smart applications for walking/biking

Professional (e.g., attorney, engineer)

66

10.1

Yes

421

64.7

Business owner/self-employed

44

6.8

No

230

35.3

Service worker

73

11.2

Used types of smart applications for walking/biking

Office/administrative/clerical worker

235

36.0

GPS/Maps (e.g., tracker, route)

242

37.2

Civil servant (government)

29

4.5

Fitness (e.g., calorie counting)

173

26.6

Home maker

76

11.7

Counter (e.g., step or distance measurement)

285

43.8

Student

35

5.4

Heart rate (e.g., pulse measurement)

103

15.8

Retiree

21

3.2

Safety (e.g., CCTV location)

23

3.5

Unemployed

29

4.5

Amenity (e.g., toilet, shelter, facilities)

43

6.6

Other

43

6.6

Augmented reality apps

11

1.7

Residential area

Other

16

2.5

Seoul-metropolitan area

428

65.6

Worry about personal safety when walking/biking

Non-metropolitan area

223

34.4

Disagree

253

38.8

Duration of answering the survey

Neither agree nor disagree

137

21.0

Between 5 and 533.8 minutes

651

100.0

Agree

261

40.2

Riding e-bike

Companions when walking/biking

Yes, I mostly ride electric bikes

27

4.1

Alone

386

59.3

No, I mostly ride conventional bikes

624

95.9

Friends

82

12.6

Providing thoughtful/honest answers

Family/relatives

162

24.9

Yes

651

100.0

Coworkers

19

2.9

No

0

0.0

Other

2

0.3

Note: The walking activity group has 325 cases, while the biking activity group has 326 cases.

  • Section 5 Conclusion should be section 6

RESPONSE 6: We appreciate your though review on our paper. We have re-numbered the sections so now section 5 is for the conclusion section as follows (refer to line 501):

  1. Conclusions

Thank you very much for your valuable review on our research!

Round 2

Reviewer 2 Report

Authors have done well job on revising the manuscript.

Reviewer 3 Report

The manuscript has been revised according to the reviewer's suggestions.